# Thermal dynamics and coalescence of Au$_{144}$(SR)$_{60}$ clusters from a machine-learned potential

Maryam Sabooni Asre Hazer ●[1] ✉, Sami Malola[1] & Hannu Häkkinen ●[2] ✉

Ligand-protected metal nanoclusters have gained significant attention due to their diverse applications in catalysis, bioimaging, and nanomedicine. While their ground-state electronic structure and optical properties have been extensively studied via density functional theory (DFT) methods, theoretical insights into their dynamic behavior, particularly for larger clusters, remain scarce due to the prohibitive numerical cost of long-timescale simulations using forces calculated from DFT. Here we investigate, using molecular dynamics (MD) simulations up to 0.12 μs timescale, thermal dynamics of the well-known Au$_{144}$(SR)$_{60}$ at 300-550 K using the recently parametrized atomic cluster expansion (ACE) potential, trained from DFT data for thiolate-protected gold clusters. Our findings reveal that thermal effects induce increased mobility in a layer-by-layer fashion, leading to formation of polymeric gold-thiolate units and rings which may fragment from the cluster at high temperatures. The remaining smaller clusters resemble experimentally observed cluster compositions. Close interaction of two Au$_{144}$(SR)$_{60}$ clusters leads to coalescence, resulting in a cluster composition and structure of the inner metal core close to ones identified in previous experiments. This work reveals mechanisms for thermal effects in ligand-protected gold clusters and larger nanoparticles that are instrumental for understanding their catalytic activity and inter-particle reactions at the atomistic level.

Monolayer-protected metal clusters (MPCs) are hybrid nanomaterials consisting of a metal core and a protecting shell of organic ligands. MPCs exhibit the electronic, optical, catalytic and biocompatible properties, which are highly dependent on their atomic and electronic structures[1–4]. Experimental work over the last 20–30 years has yielded numerous precise atomic structures of MPCs with noble metals and various organic ligands, which has made it possible to investigate their size-structure-property relations in detail via computational methods. External factors such as temperature, pressure and ligand/chemical environment can induce dramatic changes in the size, shape and properties of the clusters. In particular, thermal effects can drive critical structural transformations in

MPCs, including melting, fragmentation, and coalescence, altering their stability and functional behavior. Understanding these phenomena is fundamental to advancing our knowledge of nanoscale physics and rational design of nanoclusters with tailored functionalities. However, investigations of long-term MPC dynamics at elevated temperatures have been scarce[5] due to numerical demands in accurate DFT-based simulations and lack of more efficient, reactive classical force fields.

Machine learning (ML) has become a valuable tool in computational science, offering solutions to complex problems through data-driven models[6–12]. ML-derived force fields are currently accelerating material simulations across various domains, with applications to

[1]Department of Physics, Nanoscience Center, University of Jyväskylä, Jyväskylä, Finland. [2]Departments of Physics and Chemistry, Nanoscience Center, University of Jyväskylä, Jyväskylä, Finland. ✉e-mail: maryam.a.sabooni@jyu.fi; hannu.j.hakkinen@jyu.fi

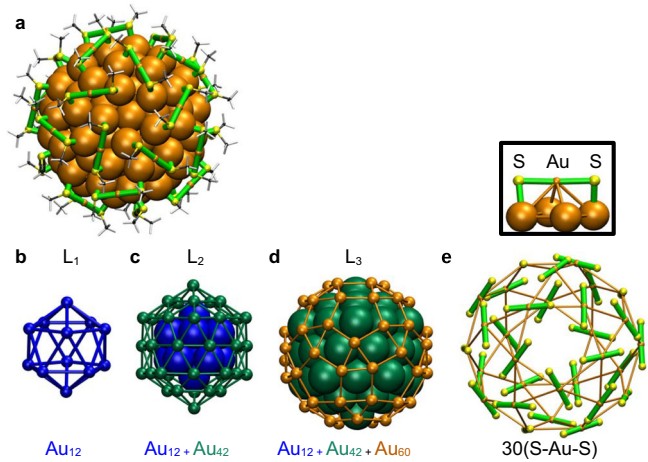

**Fig. 1 | Hierarchical structure of Au₁₄₄(SCH₃)₆₀ nanocluster.** DFT-optimized structure of $Au_{144}(SCH_3)_{60}$ (**a**), and its three core layers ($L_1$-$L_3$) shown in isolation. **b** The first inner core layer ($L_1$), consisting of 12 gold atoms in an icosahedral arrangement (blue spheres, labeled $Au_{12}$). **c** $L_2$ comprising 42 gold atoms in an icosahedral arrangement (green), shown together with the $L_1$ (labeled $Au_{12}+Au_{42}$). **d** $L_3$ forming a rhombicosidodecahedron with 60 gold atoms (orange, labeled $Au_{12}+Au_{42}+Au_{60}$). **e** Includes 30 gold atoms (orange) as part of the (SR-Au-SR) staples. The inset of **e** shows a single (SR-Au-SR) staple with sulfur atoms in yellow and the S-Au-S bonds shown in green. Methyl groups are excluded for clarity. Source data are provided as a Source Data file.

MPCs enabling structural optimizations and MD simulations for thiolate-protected gold clusters[13–16]. In this work, we focus on the thermal dynamics of the Au144(SR)60 clusters, which are one of the most studied MPCs. It was initially characterized in 1990s by Whetten and collaborators[17] as an "ubiquitous 29 kDa Au:SR cluster compound" due to the ease of synthesis and its high ambient stability, but its exact chemical composition remained a mystery until Jin's and Murray's groups determined the chemical formula from high-resolution mass spectrometry in 2009[18,19]. At the same time, DFT calculations[20] by Lopez-Acevedo et al. predicted its atomic structure (Fig. 1) that was experimentally confirmed nine years later by Wu and collaborators via X-ray single-crystal diffraction[21].

Unlike smaller MPCs, the stability of which can often be selected by considering electronic "magic numbers" (electron shell closures of superatoms)[3,22–24], $Au_{144}(SR)_{60}$ does not conform to an electronic shell closing with 84 superatom electrons. Instead, its chemical stability is assigned to an icosahedral metal core and an extended metal framework from the protected gold-thiolate units. Other studies have also explored the bonding hierarchy and ligand effects in $Au_{144}$. Yamazoe et al.[25] revealed hierarchical bond stiffness within the cluster, showing that interior Au−Au bonds are stiffer than those at the surface, in line with the icosahedral core-shell organization. This structural integrity was further confirmed when Lei et al.[26] demonstrated that replacing thiolate ligands with alkynyl groups still preserved the icosahedral $Au_{114}$ core, underscoring the structural resilience of $Au_{144}$ across ligand modifications. The electronic and optical properties of $Au_{144}$ distinguish it from smaller nanoclusters. Knappenberger et al.[27] studied its electronic relaxation dynamics by femtosecond spectroscopy and observed plasmon-like behavior in the metal core, marking the transition from molecular-like to metallic properties, in agreement with the theoretical studies by Malola et al.[28]. Kaappa et al. have pointed out that the symmetry of the metal core plays a crucial role in the optical response for clusters in this size range, where molecular-to-metallic transition takes place[29]. The chemical reactivity of $Au_{144}$ has been studied as well, particularly in ligand exchange and cluster growth. Ghosh et al.[30] found that bidentate ligands promote cluster aggregation, suggesting that ligand chemistry dictates the stability and fusion behavior of $Au_{144}$. Meanwhile,

Wijesinghe and Dass[31] demonstrated that ligand-mediated coalescence can drive $Au_{144}$ to grow into $Au_{279}$, a process likely influenced by surface mobility and ligand-induced restructuring. These properties make $Au_{144}(SR)_{60}$ an ideal model system for investigating how thermal dynamics influence the cluster stability, surface atom diffusion, ligand mobility, and the fundamental mechanisms governing structural rearrangements, fragmentation, and coalescence.

In this work, we explore these phenomena in detail, providing insights into the dynamic behavior of $Au_{144}(SCH_3)_{60}$ under varying thermal conditions. Here, we employ the ML-driven ACE potential[15] to model interatomic interactions within $Au_{144}(SR)_{60}$ cluster using MD simulations. We highlight the ability of the ACE force field to understand the dynamical and structural behavior at elevated temperatures and longer time scales of up to 0.12 µs. Furthermore, we observe and explain the coalescence process of two $Au_{144}(SR)_{60}$ clusters, producing a realistic chemical composition and metal core symmetry of a larger cluster, similar to a compound that has been identified form earlier experiments.

## Results

$Au_{144}(SR)_{60}$ clusters exhibit a hierarchical shell-like structure, as illustrated in Fig. 1a–e. The DFT-relaxed structure of $Au_{144}(SCH_3)_{60}$ is shown in Fig. 1a. At the innermost core of the cluster, there is an icosahedral $Au_{12}$ shell ($L_1$) (blue) as shown in Fig. 1b. The second icosahedral layer ($L_2$), composed of 42 gold atoms, is depicted in green and forms a Mackay icosahedral structure with $L_1$, $Au_{54}$ ($Au_{12} + Au_{42}$), as shown in Fig. 1c. The third layer ($L_3$), consists of 60 gold atoms (orange), completing the $Au_{114}$ golden cage ($Au_{12} + Au_{42} + Au_{60}$) as shown in Fig. 1d. This robust core is further stabilized by 30 monomeric staple motifs (-SR-Au-SR-) arranged in a rhombicosidodecahedral geometry as illustrated in Fig. 1e, with an inset highlighting the individual monomeric staple motif with sulfur atoms shown in yellow. The MD simulations using the ACE potential provide a comprehensive analysis of the structural stability, fragmentation dynamics, intercluster interactions, and cluster-cluster dynamics of $Au_{144}(SR)_{60}$. In this work, we studied the cluster across temperatures from 300 K to 450 K, with simulations running up to 120 ns. We collected data from four replicas, each running for 120 ns, for each temperature. The replicas at a given temperature were started from an identical atomic structure but from different initial velocities of atoms, which drives the replica runs to explore different areas of the phase space, improving statistical sampling. The replicas are labeled as I–IV. Cluster-cluster interactions were further studied through simulations at temperatures ranging from 400 K to 550 K.

A number of snapshots from the MD simulations of the single cluster with the ACE potential were selected, and the corresponding DFT energies were calculated. The correlation between the ACE and DFT energies is shown in Fig. 2 where the data is color-coded for temperatures ranging from 300 K to 450 K. The snapshots in this plot were taken from the first replica (I) of the four conducted simulations at each temperature. The analysis reveals a strong linear correlation ($R^2 = 0.9362$) between ACE-derived and DFT-calculated energies, further confirming the good performance of the ACE potential even at high temperatures.

### Structural dynamics of Au₁₄₄(SCH₃)₆₀

Figure 3 presents the pair correlation functions $g(r)$ for Au-Au and Au-S interactions at different temperatures, ranging from 300 K to 450 K averaged over all the four replicas at each temperature. The first dominant Au-Au peak appears at ≈2.97 Å, corresponding to the nearest-neighbor Au-Au bond lengths, and remains a distinct feature across all the temperatures. The first Au-S peak is positioned at a shorter distance of ≈2.3 Å. The green vertical dashed lines at the first two peaks of each Au-Au and Au-S bond distribution mark the position of the first major $g(r)$ peak, serving as a visual guide to track shifts in

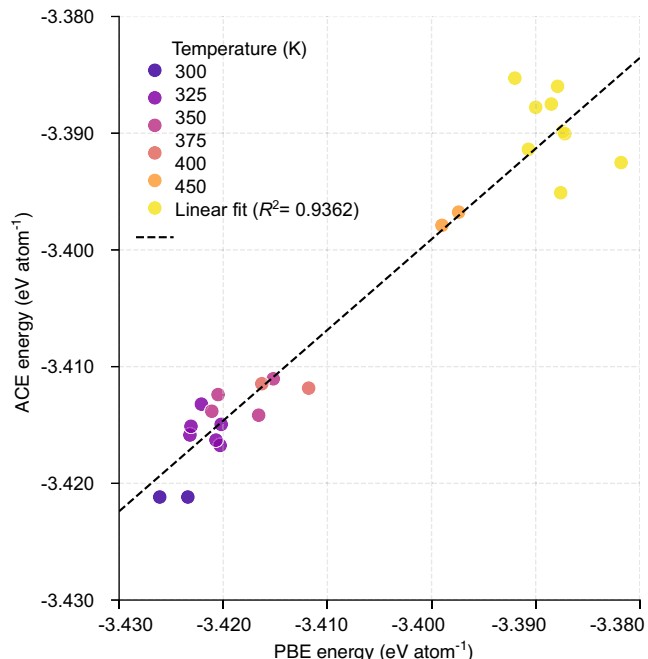

**Fig. 2 | Performance validation of ACE potential for Au₁₄₄(SR)₆₀ cluster against DFT calculations at various temperatures.** Correlation between ACE and DFT energy values per atom, for selected structures obtained from separate first replica (I) of the MD simulations at 300 K (I), 325 K (I), 350 K (I), 375 K (I), 400 K (I), and 450 K (I). The data points are color-coded for each temperature. Source data are provided as a Source Data file.

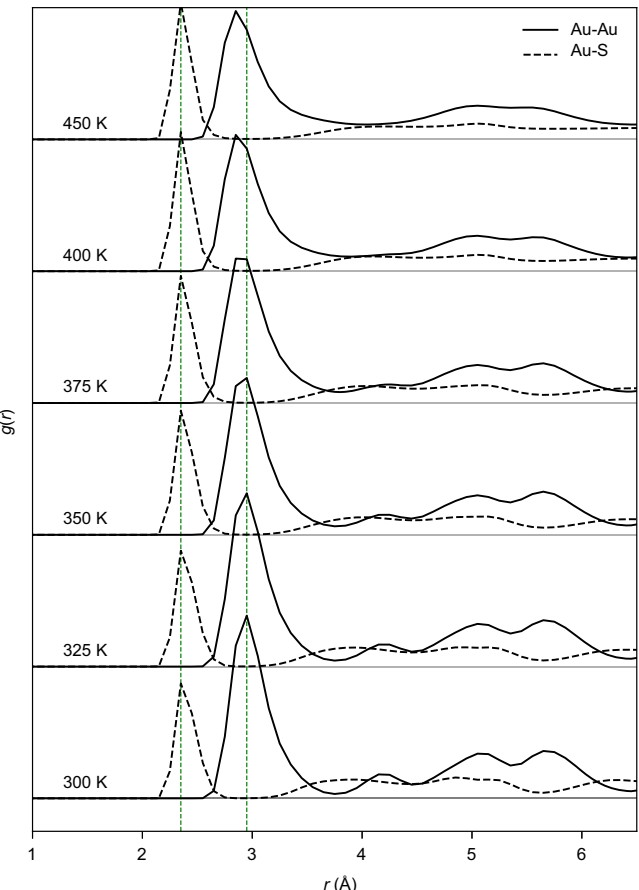

**Fig. 3 | Temperature-dependent g(r) analysis for Au–Au and Au–S bond distances in Au₁₄₄(SCH₃)₆₀ nanocluster.** g(r) analysis at temperatures from 300 to 450 K, encompassing all four replicas (I–IV). Green vertical dashed lines mark the positions of the first major g(r) peak, serving as a visual guide to track shifts in the bond length distribution by increasing temperature. The g(r) calculations account for all Au and S atoms in the simulations, including those in fragments that are partially or fully detached from the cluster at higher temperatures. Source data are provided as a Source Data file.

the bond length distribution with increasing temperature. At lower temperatures (300–350 K), the g(r) profiles exhibit well-defined, sharp peaks, indicating structurally ordered and more stable clusters. As the temperature reaches 375–400 K, the Au–Au g(r) peaks broaden, and their maxima shift slightly towards shorter bond distances. This trend becomes even more pronounced at 450 K, with a significant shift in peak maxima observed.

Figure 4a–f shows a layer-by-layer average root mean square deviation (RMSD) for gold atoms as a function of the simulation time at six temperatures from 300 K to 450 K. At each temperature, the data is combined from all four replicas (I–IV). Supplementary Figs. 1 and 2 present the RMSD data of the four individual replicas for each of the six temperatures. Moreover, Supplementary Figs. 3–26 provide detailed insights into each MD simulation replica, showing time evolution of RMSDs of each replica run and selected snapshots with time stamps visualizing the atom dynamics.

At 300 K (Fig. 4a), the average RMSD values are relatively low, reflecting primarily the vibrational motion of gold atoms. The average RMSD values for the layers are ≈0.12–0.24 Å for the 1st layer, 0.18–0.28 Å for the 2nd layer, 0.23–0.41 Å for the 3rd layer, and 0.26-0.74 Å for the 4th layer. Interestingly, the data shows systematic stepwise increase of RMSD for the 4th layer gold atoms, which originates from two of the four replica runs (Supplementary Figs. 3 and 4). Analysis of atomic configurations shows that the stepwise increase is induced by formation of elongated ligand motifs on the surface that however remain bound to the cluster.

As the temperature increases, the average RMSD rises gradually at 325 K (Fig. 4b) and 350 K (Fig. 4c) for all layers. This trend is consistently observed across the four replicas, as shown in Supplementary Fig. 1e–h and Supplementary Fig. 1i–l for 325 K (I–IV) and 350 K (I–IV), respectively. The increase in RMSD at 325 K and 350 K compared to 300 K is attributed to structural changes, as illustrated by the selected snapshots in Supplementary Figs. 7b–10b for 325 K (I–IV) and Supplementary Figs. 11b–14b for 350 K (I–IV). These changes are due to the

formation of longer gold-thiolate chains (2–3 units) that are still bound to the surface as visualized in the corresponding snapshots. Interestingly, we also observe formation of a gold-thiolate ring in three replica runs at 350 K (Supplementary Figs. 12–14). These processes at the cluster surface induce increased mobility of gold atoms also in deeper layers, leading to exchange of gold atoms between the layers.

As the temperature increases to 375 K and higher, the average RMSD, as shown in Fig. 4d–f for 375 K, 400 K, and 450 K, respectively, exhibits a significant increase and qualitative change as compared to the average RMSD calculated for 300 K, 325 K, and 350 K (Fig. 4a–c, respectively). The RMSD values for the 4th gold layer fluctuate in the range of several tens of Å, and values for the deeper layers show a steady increase during the simulation time, qualitatively indicating diffusive or "liquid-like" behavior. The high values and fluctuations for the 4th layer (in the range of 16–56 Å) are due to fragmentation of gold-thiolate chains or rings from the cluster, as visualized in Supplementary Figs. 15–26. In some cases, the fragments can return to bind back to the cluster surface, which decreases the RMSD momentarily. The surface of the cluster in these runs remains quite open and less stabilized than the thiolates, which give rise to a significant increase in mobility of atoms in the deeper layers. In some cases, we have been able to follow the diffusive motion of gold atoms from the 1st layer to the surface during one replica run. We may note that at the highest temperature of 450K, gold atoms, even (originally) in the first layer,

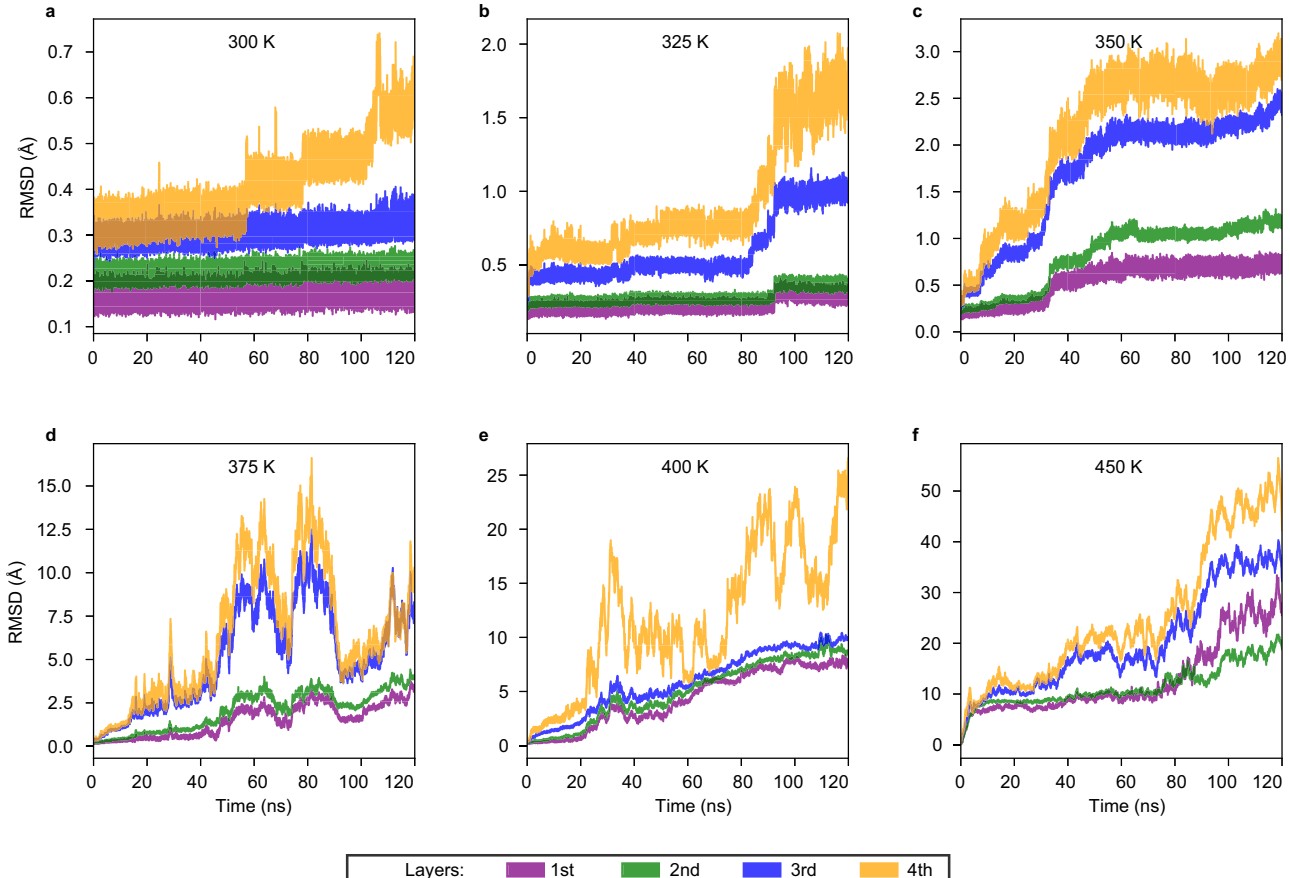

**Fig. 4 | Average RMSD of $Au_{144}(SCH_3)_{60}$ for all gold atoms in four layers, as a function of simulation time (ns).** Each layer represented by a different color: the 1st, 2nd, and 3rd layers are shown in purple, green, and blue, respectively, and the gold atoms in the ligand shell (referred to as the 4th layer) are depicted in orange. The average RMSD analysis is performed from four replicas (I–IV) at 300 K (**a**), 325 K (**b**), 350 K (**c**), 375 K (**d**), 400 K (**e**), 450 K (**f**) each for 120 ns. Source data are provided as a Source Data file.

reach the RMSD value exceeding 10 Å, which means that they have traveled (diffused) at least one cluster diameter.

Table 1 provides an overview of the fragmentation events observed at 375 K (I–IV), 400 K (II–IV), and 450 K (II–IV), highlighting the detached fragments and the molecular compositions of the remaining clusters. We observe two types of fragments, ring-like moieties in the range of $Au_4(SCH_3)_4$ to $Au_7(SCH_3)_7$ and open chains in the range of $Au_2(SCH_3)_3$ to $Au_4(SCH_3)_5$. In this data, we observe up to three fragments detaching from the cluster. We did not investigate the structure of the remaining clusters $Au_{132-142}(SR)_{47-57}$ in more detail, but we note here that experiments have identified a few stable compositions in this size range, such as $Au_{137}(SR)_{56}$[32,33] and $Au_{133}(SR)_{52}$[34].

One of the main applications of metal nanoclusters is in catalytic reactions, where their efficiency depends on the availability of surface-active sites for molecular adsorption and activation. A common method to activate a ligand-stabilized gold nanocluster to function as a catalyst is by thermal treatment to 150–250 °C[35,36], which coincides with the temperature range in our simulations. Therefore, it is interesting to analyze from our MD data how the temperature-driven transformations of the surface Au-ligand structure expose bare gold sites. We performed this analysis using the solvent-accessible surface area (SASA) method, which maps the contact area between a spherical probe of a chosen radius and the gold atoms in the cluster. Figure 5 presents the SASA results calculated for various probe sphere sizes (1.0, 1.2, and 1.4 Å) across temperatures ranging from 300 to 450 K. These results, derived from selected snapshots taken of the first replica run at different temperatures, show increased solvent exposure at

higher temperatures due to enhanced atomic fluctuations and conformational changes.

Figure 6 illustrates the solvent-accessible regions at the cluster surface for various snapshots. The snapshots were chosen to be representative of four distinct structural deformations: (i) partial opening of the ligand shell on the surface selected from 300 K (I) and shown in Fig. 6a, (ii) elongated polymer-like gold-thiolate units selected from 450 K (I) and shown in Fig. 6b, (iii) formation of a large ring-like structure on the cluster surface selected from 400 K (I) and shown in Fig. 6c, and (iv) detachment of a fragment from the cluster selected from 375 K (I) and shown in Fig. 6d. For clarity, each structure is displayed from two different orientations.

## Cluster–cluster interaction and coalescence

Understanding the atomistic mechanisms of coalescence is essential for guiding the rational design of nanomaterials for applications in catalysis, optoelectronics, and nanoscale device engineering, to name a few. In this section, we discuss a series of MD simulations at 400–550 K where two $Au_{144}(SCH_3)_{60}$ clusters were initially positioned either 3.08 or 1.6 Å (measured as the shortest distance between the outermost H atoms of each cluster) apart from each other, aiming to investigate their propensity to coalesce.

Figure 7a–k presents a series of snapshots depicting the structural evolution and interactions between the two $Au_{144}(SCH_3)_{60}$ nanoclusters during MD simulations at 400 K. The clusters are color-coded for clarity, with Au atoms in purple and S atoms in blue for one cluster, and Au atoms in red and S atoms in yellow for the other. The selected snapshots are color-coded in red and purple for clear distinction of

**Table 1 | Temperature-induced fragmentation of $Au_{144}(SCH_3)_{60}$ nanocluster**

| Temperature (K) | Detached fragments | Remained composition |
|---|---|---|
| 375 (I) | $Au_5(SCH_3)_5$ | $Au_{139}(SCH_3)_{55}$:$Au_{110}$ + $Au_{29}(SCH_3)_{55}$ |
| 375 (II) | $Au_2(SCH_3)_3$ | $Au_{142}(SCH_3)_{57}$:$Au_{114}$ + $Au_{28}(SCH_3)_{57}$ |
| 375 (III) | $Au_5(SCH_3)_5$ | $Au_{139}(SCH_3)_{55}$:$Au_{110}$ + $Au_{29}(SCH_3)_{55}$ |
| 375 (IV) | $Au_8(SCH_3)_9$ | $Au_{136}(SCH_3)_{51}$:$Au_{111}$ + $Au_{25}(SCH_3)_{51}$ |
|  | $Au_4(SCH_3)_5$ | $Au_{140}(SCH_3)_{55}$:$Au_{113}$ + $Au_{27}(SCH_3)_{55}$ |
|  | $Au_4(SCH_3)_4$ | $Au_{140}(SCH_3)_{56}$:$Au_{112}$ + $Au_{28}(SCH_3)_{56}$ |
|  | $Au_4(SCH_3)_4$ + $Au_4(SCH_3)_5$ | $Au_{136}(SCH_3)_{51}$:$Au_{111}$ + $Au_{25}(SCH_3)_{51}$ |
| 400 (II) | $Au_5(SCH_3)_5$ | $Au_{139}(SCH_3)_{55}$:$Au_{110}$ + $Au_{29}(SCH_3)_{55}$ |
| 400 (III) | $Au_4(SCH_3)_4$ | $Au_{140}(SCH_3)_{56}$:$Au_{112}$ + $Au_{28}(SCH_3)_{56}$ |
| 400 (IV) | $Au_6(SCH_3)_6$ | $Au_{138}(SCH_3)_{54}$:$Au_{111}$ + $Au_{27}(SCH_3)_{54}$ |
| 450 (II) | $Au_3(SCH_3)_4$ | $Au_{141}(SCH_3)_{56}$:$Au_{106}$ + $Au_{35}(SCH_3)_{56}$ |
|  | $Au_3(SCH_3)_4$ + $Au_5(SCH_3)_5$ | $Au_{136}(SCH_3)_{51}$:$Au_{103}$ + $Au_{33}(SCH_3)_{51}$ |
|  | $Au_3(SCH_3)_4$ + $Au_4(SCH_3)_4$ + $Au_5(SCH_3)_5$ | $Au_{132}(SCH_3)_{47}$:$Au_{107}$ + $Au_{25}(SCH_3)_{47}$ |
| 450 (III) | $Au_7(SCH_3)_7$ | $Au_{137}(SCH_3)_{53}$:$Au_{100}$ + $Au_{37}(SCH_3)_{53}$ |
|  | $Au_5(SCH_3)_5$ + $Au_7(SCH_3)_7$ | $Au_{132}(SCH_3)_{48}$:$Au_{104}$ + $Au_{28}(SCH_3)_{48}$ |
| 450 (IV) | $Au_5(SCH_3)_5$ | $Au_{139}(SCH_3)_{55}$:$Au_{110}$ + $Au_{29}(SCH_3)_{55}$ |
|  | $Au_6(SCH_3)_6$ | $Au_{138}(SCH_3)_{54}$:$Au_{111}$ + $Au_{27}(SCH_3)_{54}$ |
|  | $Au_5(SCH_3)_5$ + $Au_6(SCH_3)_6$ | $Au_{133}(SCH_3)_{49}$:$Au_{109}$ + $Au_{24}(SCH_3)_{49}$ |

Fragmentation patterns observed across four MD replicas (I–IV) at 375 K, 400 K, and 450 K. The table lists the detached fragments with the molecular compositions of the remaining clusters.

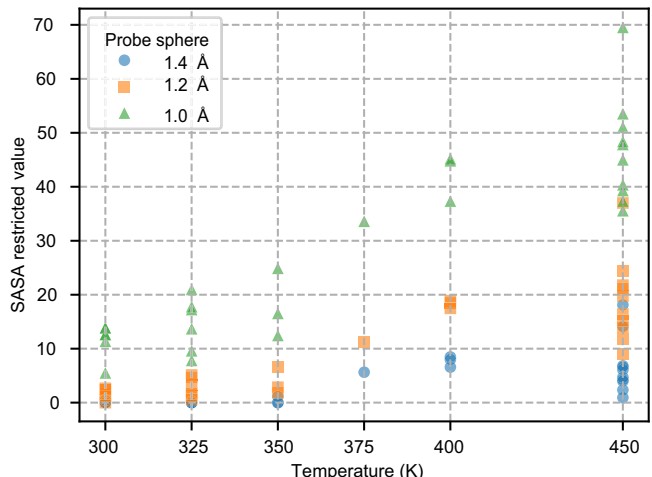

**Fig. 5 | Temperature-dependent SASA analysis for $Au_{144}(SCH_3)_{60}$ nanocluster with various probe radii.** SASA values for selected structures obtained from the first replica of MD simulations conducted at temperatures ranging from 300 K (I) to 450 K (I). The calculations were performed using probe spheres of varying radii: 1.4 Å (blue circles), 1.2 Å (orange squares), and 1.0 Å (green triangles). Source data are provided as a Source Data file.

more detailed local atomistic-level processes. Figure 7a presents the initial configuration at 0 ns, where the two clusters are positioned 3.08 Å apart. As the simulation progresses, Fig. 7b illustrates the early thermal fluctuations, where some ligands begin to elongate slightly, but the clusters remain structurally relatively intact. Moving to Fig. 7c at ≈12 ns, ligand dynamics become more pronounced as several thiolate ligands extend outward, forming ring-like structures that remain attached to the cluster surface. By Fig. 7d at 17.40 ns, ligand-mediated bridging between the clusters initiates. Simultaneously, a second structural fragment, an $Au_4(SCH_3)_4$, detaches from the red-colored cluster. From Fig. 7d at 17.40 ns onward, the number and composition of the fragmented structures remain consistent, with no change in their atom counts across subsequent frames. These fragments exhibit

dynamic movement around the clusters, suggesting that the system reaches an approximate dynamic equilibrium where the fragments continuously move without undergoing further structural decomposition or aggregation. Between Fig. 7e at 47.68 ns and Fig. 7f at 50.73 ns, the ligand bridge extended. By the end of the simulation at ≈88 ns, the clusters maintain their partially connected state without undergoing full coalescence. The formation of an inter-cluster bridge at 400 K without complete coalescence can be attributed to the behavior of nanoparticles as governed by their size and atomic coordination[37–40]. The relatively large size of Au144 results in a significant fraction of interior atoms with high coordination numbers (CN), reducing their reactivity compared to surface atoms. In contrast, surface atoms with lower CN values exhibit higher mobility, facilitating ligand-shell restructuring and the formation of molecular bridges under elevated thermal conditions. However, at 400 K, the energy barrier for complete coalescence is not fully overcome. Instead, the system undergoes partial restructuring, where surface atoms migrate, form transient bridges, or contribute to ligand-shell fragmentation. The lack of full coalescence at 400 K contrasts with the behavior of smaller clusters, where a higher surface-to-volume ratio results in a greater proportion of low-CN atoms, enhancing their reactivity. Our previous study of coalescence of smaller $Au_{25}(SR)_{18}$ clusters[15] demonstrated that, at similar temperatures, these systems undergo more rapid fusion, as the energetic barriers for atomic diffusion and coalescence are likely lower. Further MD simulations were performed at 500 K (Supplementary Fig. 27) and 550 K (Supplementary Fig. 28), with two $Au_{144}(SCH_3)_{60}$ clusters initially positioned 3.08 Å apart to assess the critical temperature necessary for coalescence. However, no complete merging was observed under these conditions. Since no coalescence was observed during these simulations, we reduced the initial cluster–cluster distance to 1.6 Å and ran simulations at 500 K and 550 K. Although reducing the cluster separation to 500 K, as shown in Supplementary Fig. 29 did not lead to coalescence, fragmentation was observed. However, when the temperature was increased to 550 K with the reduced distance, coalescence was achieved, as illustrated in Fig. 8a–i.

Supplementary Table 1 provides a summary of the fragmentation events observed during the simulations of two $Au_{144}(SR)_{60}$ clusters at

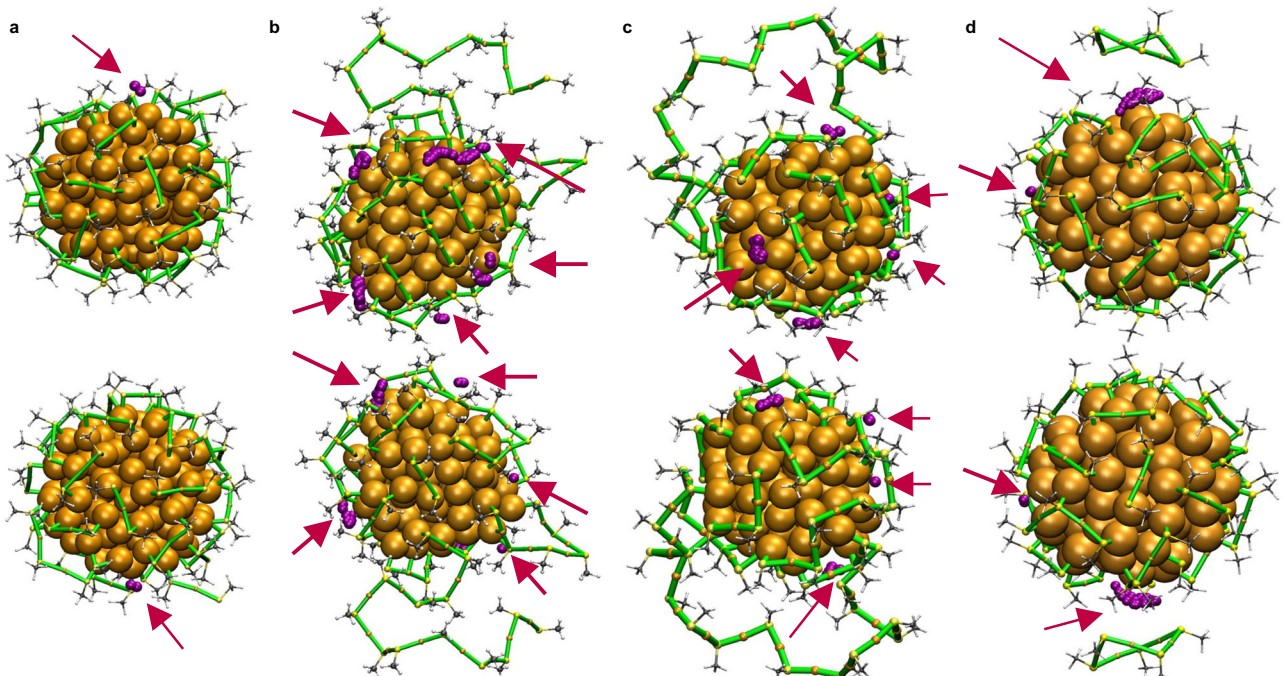

**Fig. 6 | Visualization of the SASA analysis for selected structures obtained from MD simulations.** The purple spheres represent the positions of adsorbates, modeled as spheres with a radius of 1.4 Å, in contact with the core Au atoms of the cluster. **a**–**d** show selected snapshots from the first replica (I) at: 300 K (I), 450 K (I), and 400 K (I), and 375 K (I), respectively. Each structure is shown from two different orientations to enhance clarity. Source data are provided as a Source Data file.

different temperatures. At 400 K, with a cluster separation of 3.08 Å and a simulation duration of 87.60 ns, limited fragmentation was observed, resulting in the detachment of only two distinct fragments: one $Au_7(SR)_7$ and one $Au_4(SR)_4$ complex. In contrast, the simulation at 550 K, with a cluster separation of 1.6 Å over 109.33 ns, exhibited significantly more fragmentation. This simulation produced a diverse array of fragments, including single fragments of $Au_8(SR)_8$, $Au_6(SR)_6$, $Au_3(SR)_4$, and $Au(SR)_2$. Furthermore, we observed recurring detachments of certain fragments, $Au_5(SR)_5$ separated from the main cluster, which occurred three times on separate occasions, while $Au_4(SR)_4$ fragments appeared four times throughout the simulation.

Figure 8a–i displays a series of snapshots capturing the coalescence mechanism during the simulation. Figure 8a shows the two clusters at the beginning of the simulation, positioned 1.6 Å apart. As the simulation progresses, Fig. 8b at 2.39 ns shows atomic fluctuations, with the outer shells beginning to deform, while the core Au atoms remain largely intact. Furthermore, fragmentation occurs, resulting in the formation of $Au_4(SR)_4$ and $Au_3(SR)_4$. Incorporation of core Au atoms in the fragment is observed in the $Au_3(SR)_4$ structure. Throughout the rest of the simulation, fragments that are far from the remaining clusters are not depicted in the snapshots; however, a complete list of all fragments is provided in Supplementary Table 1. By Fig. 8c at 14.20 ns, the clusters show increased interaction, with atoms from the outer shells beginning to form a bridge. This bridge not only includes Au atoms from the ligand staples but also incorporates core Au atoms (as indicated by the difference in their sphere sizes), which have migrated to this region and contribute to the bridge formation. In Fig. 8d at 21.52 ns, more fragmentation is observed, along with the first instance of core Au atoms migration from the cluster in purple to the second cluster. This migration is pointed out in Fig. 8d using blue dashed lines. Figure 8e–h illustrates the progressive stages of cluster merging. In Fig. 8e at 24.13 ns, the two clusters move closer together, followed by further integration in Fig. 8f at 24.92 ns. An atom string, consisting of both purple and red Au atoms along with yellow and green

S atoms from each cluster, becomes visible. By the final snapshot in Fig. 8i at 109.33 ns, a short atom string, incorporating atoms from both clusters, can still be observed. The merged cluster in Fig. 8i adopts a nearly spherical shape with a stoichiometry of $Au_{239}(SR)_{69}$, consisting of 197 core Au atoms and 42 Au atoms within the $Au_{42}(SR)_{69}$ found at the various staple motifs. The optimized $Au_{239}(SR)_{69}$ cluster displays an approximately ellipsoidal gold core with a more compact gold core and Au-S interface region. Our neighbor analysis, shown in Supplementary Fig. 30 confirmed the absence of any sulfur atoms (hence thiols) trapped inside the gold core during the coalescence process.

It is very interesting to note that the molecular composition of $Au_{239}(SR)_{69}$ falls closely within the estimated composition of $Au_{210-230}(p\text{-mercaptobenzoic acid }(p\text{MBA}))_{70-80}$, identified in a 2019 study[41] reporting a synthesis of a plasmonic, water-soluble gold cluster. Although our merged cluster is a "product" of a completely different chemical process as compared to room-temperature, pH-controlled synthesis of a water-soluble gold cluster, the similar stoichiometry strongly suggests that the ACE potential is able to maintain a very realistic ligand density at the cluster surface after coalescence, by ejecting surplus material as gold-thiolate fragments out of the system.

To study further the structure of the $Au_{239}(SR)_{69}$ cluster, we started from the last configuration from the 550 K MD run, and step-wise cooled it down to 300 K with 50 K steps (13.33 ns for each step), and finally optimized the cluster by using the ACE potential. The analysis of the optimized structure is shown in Fig. 9. We find an inner metal core (50 gold atoms) to have a twinned FCC structure, with the surface region having multiple gold-thiolate units $Au_n(SR)_{n+1}$ ($n = 1...6$) and one additional $Au_9(SR)_9$ moiety (Fig. 9b, c). We modeled the powder X-ray diffraction (PXRD) pattern of the structure (see "Methods") and compared it to the measured PXRD data for the cluster $Au_{210-230}(p\text{MBA})_{70-80}$, synthesized in ref. 41. As Fig. 9a shows, the computed PXRD pattern agrees quite well with the measured data for $Au_{210-230}(p\text{MBA})_{70-80}$, which was also determined to have the twinned

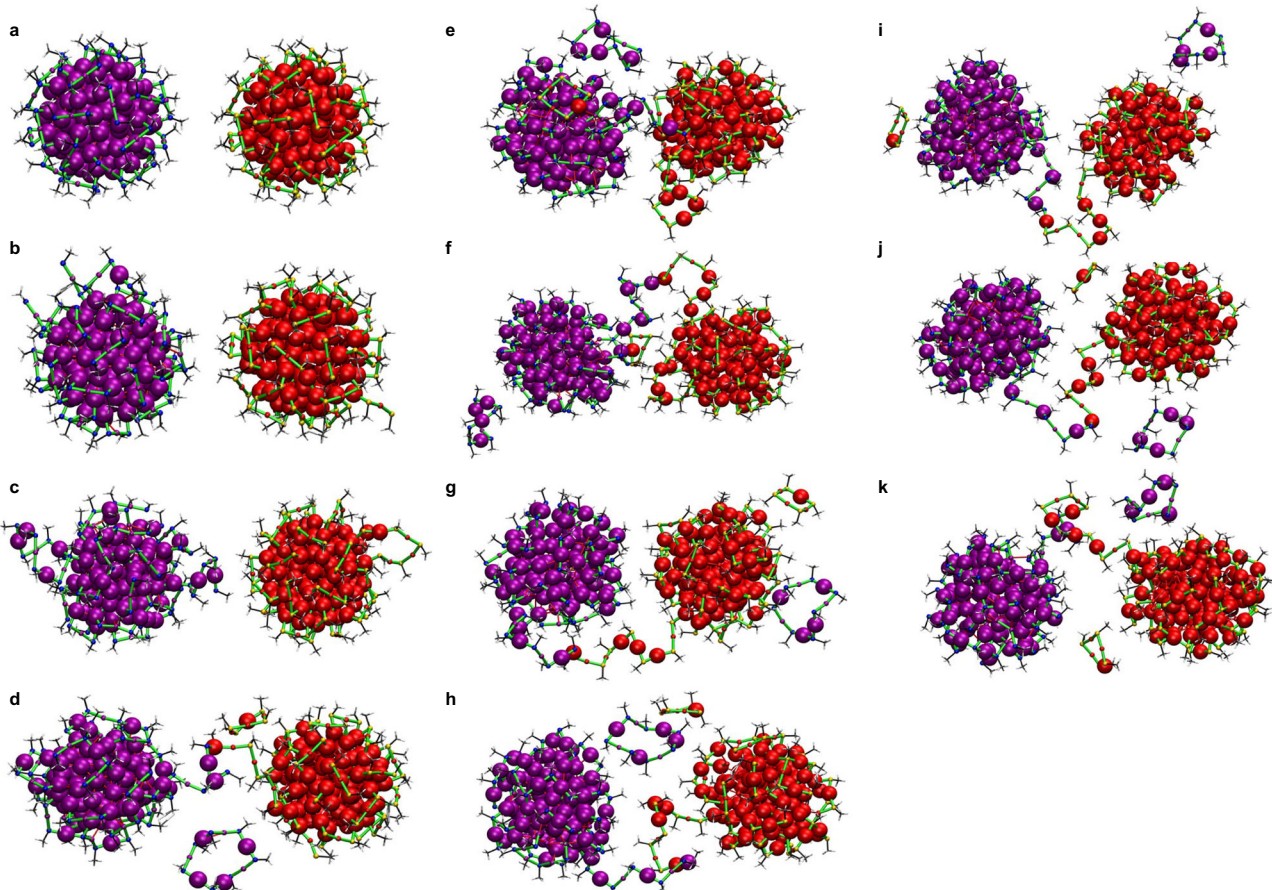

**Fig. 7 | Fusion dynamics of two $Au_{144}(SCH_3)_{60}$ nanoclusters at 400 K.** Selected snapshots shown in (**a–k**) from MD simulations of two $Au_{144}(SCH_3)_{60}$ clusters at 400 K positioned 3.08 Å apart, shown in red and purple for distinction. The snapshots are taken at: **a**: 0 ns, **b**: 0.62 ns, **c**: 11.27 ns, **d**: 17.40 ns, **e**: 47.68 ns, **f**: 50.73 ns, **g**: 55.66 ns, **h**: 64.82 ns, **i**: 66.70 ns, **j**: 74.22 ns, **k**: 87.60 ns. Core Au atoms are represented as the largest spheres, Au atoms in ligand staples and sulfur atoms are shown with the smallest spheres. Each orientation has been adjusted to provide a better perspective of the structural dynamics. Source data are provided as a Source Data file.

FCC metal core as the most likely structure[41]. Thus, the ACE potential is able not only to produce a merged larger cluster from two $Au_{144}(SR)_{60}$ clusters with a realistic chemical composition but also with a realistic metal core symmetry.

## Discussion

We have applied a recently developed machine-learned atomistic potential (ACE) to investigate thermal dynamics of the well-known $Au_{144}(SR)_{60}$ cluster at 300–550 K. Its electronic and atomic structures have been extensively studied via density functional theory methods since it was first theoretically predicted in early 2009 and experimentally proven from high-resolution mass spectrometry later in the same year. However, little has been known theoretically about the atomic-level dynamics and stability of this cluster at elevated temperatures since DFT MD calculations can only reach the picosecond time scale, and reliable, numerically efficient chemically reactive classical force fields have been missing. The ACE potential enables now reliable molecular dynamics simulations at elevated temperatures extending to 0.12 ms, about five orders of magnitude improvement in the time scale.

We find that the temperature-induced disordering of the cluster takes place via a layer-by-layer mechanism, and the outermost gold-thiolate layer, consisting of 30 RS-Au-SR units, spontaneously forms open (polymer-like) or closed (ring-like) gold-thiolate moieties. These fragments may detach and again join the cluster in a dynamic fashion. This observation is relevant

to many experimental results that have documented gold-thiolate fragments flying from the parent cluster. Chemical compositions of the remaining clusters (with 132–141 gold atoms and 47–56 thiolates) are within the experimentally reported range. Detachment of gold-thiolate units or larger fragments has also been implied from catalytic experiments, where the activation of the cluster on oxide supports is usually done via thermal treatment at temperatures relevant to our simulations.

We also investigated interactions and a reaction between two $Au_{144}(SR)_{60}$ clusters. At elevated temperatures, clusters can exchange gold atoms and thiolates by first establishing a cluster-cluster bridge from a long gold-thiolate polymer. When at close enough contact, this interaction can lead to a complete coalescence of the cluster to form a new, roughly spherical cluster. The reaction product establishes a new gold-thiolate surface layer by fragmenting similar open or closed gold-thiolate moieties from the cluster as in the case of a single $Au_{144}(SR)_{60}$ cluster. The composition of the reaction product ($Au_{239}(SR)_{69}$) is close to an earlier synthesized gold-thiolate cluster $Au_{210–230}(pMBA)_{70–80}$, implying that the ACE potential is able to maintain a realistic surface density of thiolates also at the surface of a larger cluster. Furthermore, the structure of the inner metal core has a twinned FCC symmetry, similar to the structure of the $Au_{210–230}(pMBA)_{70–80}$. We find this performance of the ACE potential quite remarkable. Our work now opens roads to detailed atomic-level simulation studies of cluster-cluster interactions and reactions, giving insight into atomistic processes behind cluster catalysis, thermal stability, fragmentation and

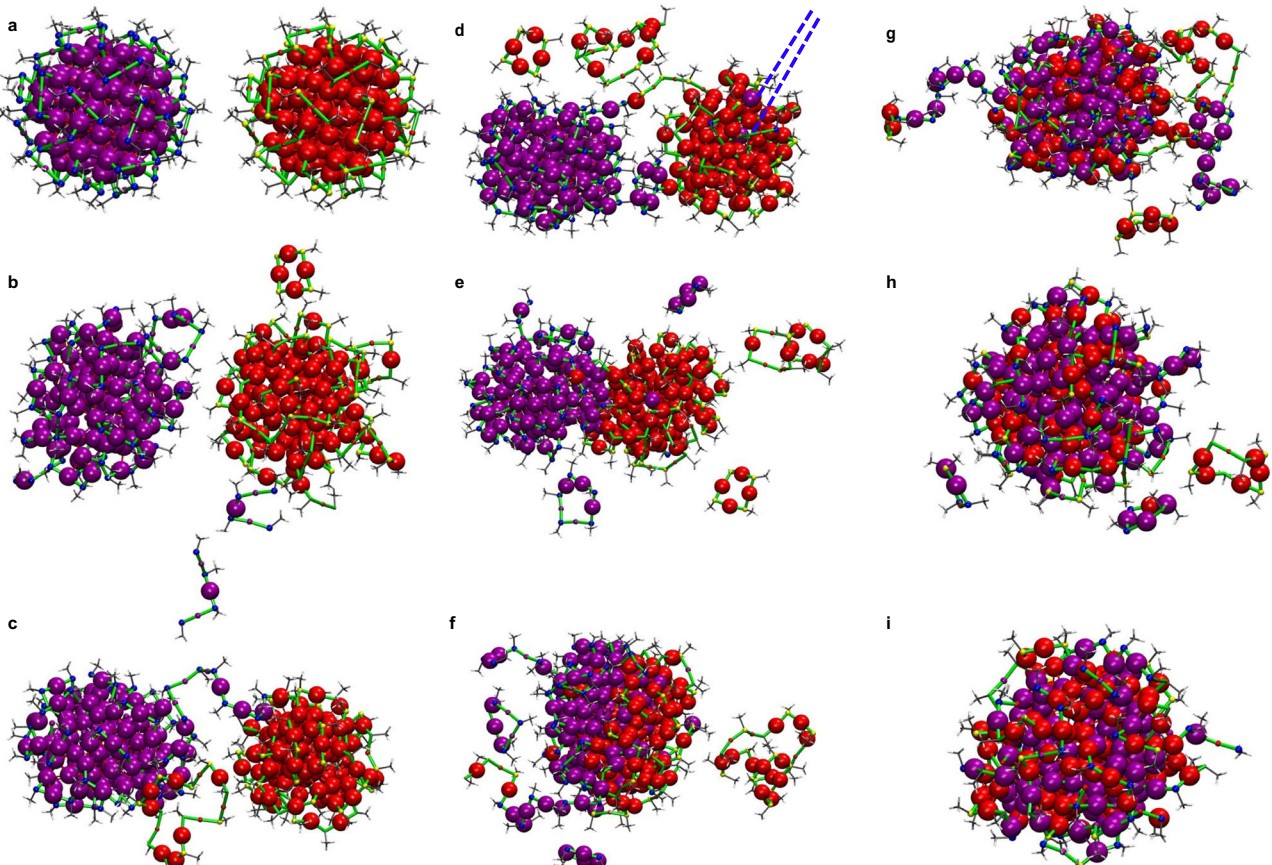

**Fig. 8 | Fusion dynamics of two Au144(SCH3)₆₀ nanoclusters at 550 K.** Selected snapshots shown in (**a**–**i**) from MD simulations of two $Au_{144}(SCH_3)_{60}$ clusters at 550 K positioned 1.6 Å apart, shown in Au in purple and S in blue for one cluster, and Au in red and S in yellow for the other. Core Au atoms are shown as the largest spheres, Au atoms in ligand staples and sulfur atoms shown with smallest spheres.

The snapshots are taken at: **a**: 0 ns, **b**: 2.39 ns, **c**: 14.20 ns, **d**: 21.52 ns, **e**: 24.13 ns, **f**: 24.92 ns, **g**: 26.88 ns, **h**: 72.98 ns, **i**: 109.33 ns. Each orientation has been adjusted to provide a better perspective of the structural dynamics. The blue dashed line shown in (**d**) indicates the first two Au atoms migrating from the purple cluster to the red cluster. Source data are provided as a Source Data file.

coalescence, applicable also to larger metal nanoparticles that are stabilized by ligand molecules. Prior experimental work has documented merging of 2-3 nm gold nanoparticles, protected by weakly binding cetyltrimethylammonium bromide (CTAB) ligands, in a graphene liquid cell via imaging by transmission electron microsopy[42]. With the progress of simulation methods demonstrated here, such processes can now be investigated in great atomistic detail.

## Methods
### ACE model
The ACE model[15] was trained on over 30,000 DFT calculations, including various packings of gold core (icosahedral, decahedral, and face-centered cubic (FCC) structures) and sizes up to $Au_{144}$. The dataset included MD trajectories and active learning sampled clusters, with 31,942 structures used for training and 11,702 for validation. DFT calculations were performed, ensuring consistency with prior studies. The trained ACE potential accurately predicted energies and atomic forces, achieving a root mean squared error of 11.0 meV per atom for the training set and 2.27 meV per atom for the testing set. MD simulations up to 0.12 μs revealed key cluster transformations, including chiral isomerization in $Au_{25}(SR)_{18}$, where chiral isomers $Au_{25}C_1$ and $Au_{25}C_2$ were identified, leading to a high-energy intermediate $Au_{25}D$ with an open facet. The study also captured intercluster reactions and coalescence mechanisms, demonstrating that two $Au_{25}(SR)_{18}$ clusters at 500 K fused through ligand exchange and rearrangement, forming transient intermediates of varying sizes $Au_{50-n}(SR)_{36-n}$. The simulations revealed the formation of $(Au-S)_n$ rings ($n = 4, 5, 6, 8, 12, 13$), which facilitated size equilibration and metal exchange. Additionally, high-temperature simulations at 450 K and 500 K showed ligand-shell rearrangements, leading to the fusion of $Au_{25}(SR)_{18}$ into larger clusters such as $Au_{38}(SR)_{24}$, aligning with experimental observations.

### MD simulations
The MD simulations were conducted using the Large-scale Atomic/Molecular Massively Parallel Simulator (LAMMPS)[43], an open-source MD code that enables modeling of atomic and coarse-grained systems using various force fields. For this study, we employed the Au-S-C-H ACE potential[15], a machine-learned interatomic potential. Recently, it was parametrized for thiolate-protected gold clusters using an extensive DFT database for energies and atomic forces over a range of cluster sizes and compositions from $Au_{18}(SR)_{14}$ to $Au_{144}(SR)_{60}$.

All MD simulations were performed using the performant atomic cluster expansion (PACE) implementation within LAMMPS (ML-PACE)[44], using the *NVT* ensemble with a Langevin thermostat[45] for temperature control. The MD timestep was 1 fs. The Langevin thermostat mimics the experimental thermalization process by introducing a stochastic force and a frictional force. The stochastic force emulates realistic thermal fluctuations, and the frictional force provides a feedback to keep the average kinetic energy of the system at a value that corresponds to the defined temperature via the classical equipartition theorem. The damping coefficient of the thermostat was 1.0 ps⁻¹. For all cases, whether simulating a single cluster or studying the coalescence of two clusters, thermalization was achieved within

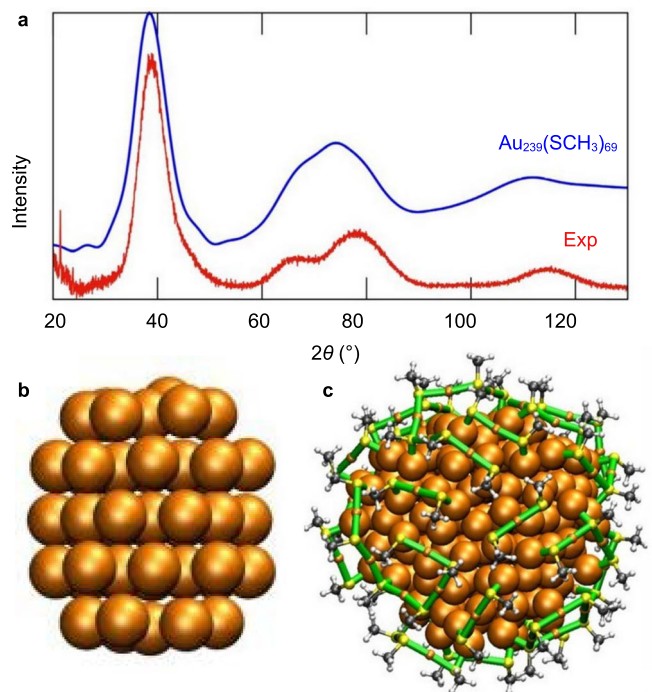

**Fig. 9 | Structural characterization of $Au_{239}(SR)_{69}$ nanocluster. a** Computed PXRD function (blue) for the optimized $Au_{239}(SR)_{69}$ cluster compared to the measured PRXD data (red) for the $Au_{210-230}$ (*p*MBA)$_{70-80}$ compound. **b** the 50-atom inner core, indicating an FCC stacking fault and **c** the full structure of the $Au_{239}(SR)_{69}$ cluster, with the identified surface gold-thiolate moieties listed. Source data are provided as a Source Data file.

9–20 ps from the start of the simulation run. To minimize the energy of the initial structure, conjugate gradient minimization was performed with strict energy and force tolerances of $10^{-4}$ and $10^{-6}$, respectively.

### RMSD analysis
The RMSD for gold atoms in layers was calculated from Eq. (1) at each time step where the structure data was saved, with reference to the initial time $t_0$. Atoms were assigned a given layer according to the initial configuration of each replica run.

$$\text{RMSD}(t) = \sqrt{\frac{\sum_{i=1}^{N_{Au}}(\mathbf{r}_i(t) - \mathbf{r}_i(t_0))^2}{N_{Au}}} \qquad (1)$$

### XRD analysis
Theoretical XRD patterns were calculated as described in ref. 20. Modeling of powder XRD curves includes the thermal damping by a Debye–Waller factor $\exp(-Bs^2/2)$ related to thermal vibrations where $s = 2\sin\theta/\lambda$, with $\lambda = 1.54$ Å and $B = 0.04$ nm$^2$. In addition, atomic numbers were used to describe the form factors for different atoms. Experimental powder XRD curve of the protected cluster with estimated composition of $Au_{210-230}(SR)_{70-80}$ was taken from ref. 41.

### Technical details of MD performance
The computational performance was assessed using a single NVIDIA Volta V100 GPU provided by the Finnish national supercomputing center CSC, achieving an average simulation speed of 83 ns per day for a single $Au_{144}(SCH_3)_{60}$ cluster and 51 ns per day for two clusters in the same computational unit cell. Our simulations employed LAMMPS with the Type Label Framework[46], using the GPU-accelerated KOKKOS implementation of PACE to optimize

computational efficiency. All structural and simulation results were visualized using VMD (1.9.4a57)[47] software.

### Reporting summary
Further information on research design is available in the Nature Portfolio Reporting Summary linked to this article.

## Data availability
The data that support the findings of this study are available from the corresponding authors upon request. Raw data from all MD simulations are provided from ref. 48. Source data are provided with this paper.

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

## Acknowledgements

This work was funded by the Research Council of Finland and by the European Research Council (ERC AdG project DYNANOINT). The computations were done at the Finnish National Supercomputing Center CSC. M.S.A.H. thanks Caitlin McCandler for useful discussions.

## Author contributions

M.S.A.H. conceived and designed the project. H.H. supervised the project. M.S.A.H. performed all MD simulations, analyzed the results, and wrote the initial draft of the paper. S.M. conducted computational XRD analysis. H.H. and S.M. commented on the manuscript drafts. All results were discussed among all authors.

## Competing interests

The authors declare no competing interests.
