## [Transparent Peer Review file · Nature Communications]

Thermal dynamics and coalescence of Au₁₄₄(SR)₆₀ clusters from a machine-learned potential

Corresponding Author: Professor Hannu Häkkinen

Version 0:

Reviewer comments:

Reviewer #1

(Remarks to the Author)

The manuscript "Thermal Dynamics and Coalescence of Au₁₄₄(SR)₆₀ clusters from a Machine-Learned Potential" is a purely theoretical molecular dynamics study of such clusters. First, the authors investigate the structural variability of the clusters as a function of temperature, and then explore how this variability may affect the potential coalescence of two clusters. The paper is mostly well written and the simulations are performed in a mostly appropriate fashion. I would recommend some methodological clarifications and further work, but I don't think these are very important. However, my main concern with respect to publication at Nature Communications is the level of novelty, or rather the lack thereof. I found the paper a technical paper that does not present any new physics or even unexpected result that would give its readers new intuition on the topic (except some incremental knowledge). As such, I cannot recommend it for publication in Nature Communications.

Hoping that my review will help the authors find a right outlet for their good work, I would also like to add a few questions/comments, which will hopefully help improve the quality of the paper:

1. The abstract does not communicate any particular importance of the work. It should be clearer why this is a high impact paper.
2. I believe the Methodology section could be more detailed. For example, what is the statistical ensemble used and how does it reflect the experimental system that is simulated? How were the particles thermalised? In the case of coalescence, were they equilibrated in advance, and, if yes, for how long? Why were the two particular distances between the particles chosen? Since there was a thermostat used, why would the authors expect that a closer distance between clusters affect their degree of coalescence? In my opinion, the authors seem to focus too much on the implementation of the ML potential (which is fine, since this constitutes a methodological advance or novelty), but not enough on the basics of MD.
3. Since velocities are appointed randomly to the atoms in an MD simulation, equilibration is important. Also, statistics. Are the RMSDs of various temperatures reproducible, or are they a result of randomness? Without any statistics we cannot be sure whether the results are physically meaningful. For example, it is not easy to believe that the different features between the 325 and 330 K simulations are down to the 5 K difference. We do have 3 cases for 450 K; why not for the other temperatures? I do not think that 3 cases are enough, by the way, especially considering that they gave so different results. The simulation setup is basic and the authors could easily run a great number of simulations without too much effort.
4. I am not 100% convinced with the analysis of the RMSD results. RMSD take averages for all atoms of each layer; therefore, if a protrusion is formed (or, especially when a part of the cluster is detached) that totally upsets the averaged RMSD values. I do not see the "migration between layers" the authors discuss about. For example, in lines 190-93: at 350 K, L2 crosses 0.8 Å and L3 approaches 1.5 Å, but I don't see any line mixing in Fig. 4. In my opinion, all features in Fig. 4 are related to protrusions, not atomic rearrangement.

I hope that the authors will take the time to work on these issues for the benefit of their already good work so far, and that my suggestions will help them publish a very good paper in another journal.

Reviewer #2

(Remarks to the Author)

Metal clusters (NCs) have attracted extensive interest in recent years. The authors selected one of the most studied, near

critical sized Au nanoclusters- Au₁₄₄(SR)₆₀ for thermal dynamics investigation at 300 K - 550 K by applying a recently developed machine-learned atomistic potential (ACE). They found that the temperature-induced disordering of the cluster takes place via layer-by-layer mechanism and the outermost gold-thiolate layer, consisting of 30 RS-Au-SR units, spontaneously forms open (polymer-like) or closed (ring-like) gold-thiolate moieties. These fragments may detach and again join the cluster in a dynamic fashion. They also investigated interactions and a reaction between two Au₁₄₄(SR)₆₀ clusters: at elevated temperatures, clusters can exchange gold atoms and thiols by first establishing a cluster-cluster bridge from a long gold-thiolate polymer; when at close enough contact, this interaction can lead to a complete coalescence of the cluster to form a new roughly spherical cluster. Overall, these findings and results are very interesting, novel and inspiring, which are mirrored by some previous experiments, therefore I can recommend the publication of this nice work after addressing the following issues:

1. Why are the durations of 120 ns, 87.27 ns, and 89.85 ns considered in this work? For realistic experiments, they last hours or even days. Does it mean that the simulation can not really reflect the realistic situation?
2. What is the MD simulation limitations(disadvantages) for metal nanocluster investigations?
3. Can the authors give a precision and efficiency (time-consuming) comparisons between AIMD method and ML-potential in term of Au₁₄₄(SR)₆₀? Which atomic characteristic does ACE model denote? Why did the authors use this model? How does it accelerate the calculation speed.?
4. The RMSD results (Fig.4) can be further clarified or interpreted. For example, it is unknown what is the electronic energy change of each sharp change in Figure 4? Can the energy change of each peak indicate the feasibility of the according reaction? etc.
5. One of the authors'opinions is that the restructuring of gold nanoclusters starts from the change of the outmost gold-thiolate layer, however, a structure transformation from Au₁₂₇ to Au₁₂₆ was very recently reported, in which the innermost gold of Au₁₂₇ was removed with the left structures including the outmost layer remained after the transformation. Basing on the fact, my question is whether there is also the possibility that the restructuring of gold nanoclusters starts from the innermost gold atom?

Reviewer #3

(Remarks to the Author)

In this work, the authors utilize the atomic cluster expansion (ACE) machine-learning potential to investigate the thermal dynamics and coalescence of Au₁₄₄(SR)₆₀ clusters. While these clusters have been extensively studied within the cluster community, research focusing on their thermal dynamics and coalescence is relatively scarce. This study employs machine learning to extend the time scales that have not previously been explored for Au₁₄₄, unveiling several intriguing findings that pave the way for a deeper understanding of cluster-cluster interactions and coalescence at the atomistic level. I recommend that the manuscript be published in Nature Communications after minor revisions.

1. The simulations utilize varying lengths of molecular dynamics (MD). It would be helpful to explain how the decision was made regarding when to terminate the simulations.
2. The authors conducted three independent simulations at 450 K. What was the original motivation for this approach, and why do the simulation times differ?
3. The caption for Figure 6 needs to be corrected.

Version 1:

Reviewer comments:

Reviewer #1

(Remarks to the Author)

I would like to thank the authors for heavily revising their original manuscript according to the suggestions by all the reviewers. However, while I appreciate their effort and understand their position, in all honesty, I still do not consider the manuscript fit to publish in Nature Communications. Let me explain:

First, unless I am still missing something important, I find the results rather expectable. In principle, we have two metallic nanoparticles whose ligand shell protects them from coalescing (which would happen spontaneously if unprotected), and when the shell is disrupted by spontaneous fragmentation, the metallic cores can get in touch and the two clusters coalesce. I am not an expert on ligand-protected nanoparticles, but I don't see what is novel here (for example, isn't this, more or less, similar to Bae et al. Nanoletters, 2020, 20, 8704?).

As I said, I am not an expert on ligand-protected nanoparticles, so let us assume that this result is, indeed, groundbreaking. I am still not sure about how convincing it is. For example, the authors obliged by performing additional simulations. Then, they present in figure 4 the average RMSD evolution of 4 simulations/temperature. Bearing in mind that the individual RMSD graphs from SI are so different to each other, I am not sure there is a point in averaging them. For example, the two spikes around 60 and 80 ns at 375 K come from simulation II, which has a very flat profile otherwise; some protruding chains are created, which soon reunite with the cluster. The slope in the average profile of figure 4d after 100 ns comes from simulation III, where such spikes were not evident. I don't see the point of averaging.

Interestingly, the SASA probing is not performed for all simulation runs.

Further, with respect to our previous discussion about diffusion (see Response Letter), I still cannot consider these protrusions "diffusion" of Au atoms to an external shell. The whole structure is totally disrupted, exploding one might say; that is not how I understand diffusion. But, again, perhaps this is just a matter of interpretation.

Can we be sure that this fragmentation is not an artifact of the inter-atomic potential or of poor relaxation/equilibration? I've had similar effects in similar simulations, and it always turned out I was doing something wrong. This is clearly something very sensitively related to the potential. Has this instability at high temperatures been confirmed by, say, DFT-MD?

Again, with respect to the statistics: the authors claim that when the two coalescing nanoparticles were brought to a higher proximity, they could coalesce (at least more probably than at longer distances). While I don't find this completely implausible, shouldn't this probability depend more on the exact configuration? For example, if two nanoparticles that had a high tendency to fragment were brought together, wouldn't they have a higher probability to coalesce fully? This is yet another claim that needs to be statistically supported further. Can the authors be sure that the same two nanoparticles that coalesced in their simulation would still coalesce if brought together at the same distance but with a different relative orientation?

I understand that my comments may be frustrating, after the authors having performed a lot of work, and especially considering that the other two reviewers gave positive assessment. However, I find it hard to accept the manuscript, on grounds of doubts about the conclusions made from the results. I trust the editor can evaluate the situation, and if they find I am being unreasonably doubtful I would be happy to hear the paper was accepted.

Reviewer #2

(Remarks to the Author)

The authors addressed my concerns (comments), but it is suggested that the authors added the according discussions in the revised manuscript to increase the readability.

Reviewer #3

(Remarks to the Author)

The authors have addressed all my comments, and I recommend that the manuscript be published as an article in Nature Communications.

Version 2:

Reviewer comments:

Reviewer #1

(Remarks to the Author)

The authors have answered all my previous comments in a somewhat satisfactory way. I fully agree with their decision not to retouch their manuscript, as this wouldn't have done it any good. Therefore, I accept the manuscript for publication.

Response to Referees

General remarks from the authors: We thank all Referees for their careful reading and insightful comments and questions. Referees 2 and 3 recommend publication in Nature Communication after a minor revision while Referee 1 is critical about the novelty and impact of the work, while assessing positively the overall quality. Below, we give a point-by-point response to all comments by the Referees, but here we first summarize the main revisions:

1. The abstract has been rewritten to address the novelty, importance and impact of this work
2. Figure 3 has been redrawn to make it visually more clear
3. Figure 4 has been redrawn based on the new improved statistical sampling of MD trajectories at all temperatures
4. Discussion of RMSD data (Figure 4) and thermal behaviour has been extensively revised.
5. Table 1 has been revised to include data for fragmentation patterns from the improved statistics
6. Methods section has been extended

For the convenience of the Referees, we have provided an auxiliary pdf manuscript file where all the major changes in the text are highlighted by blue font (file MS-Au144-marked.pdf).

Reviewer #1 (Remarks to the Author):

The manuscript "Thermal Dynamics and Coalescence of Au₁₄₄(SR)₆₀ clusters from a Machine-Learned Potential" is a purely theoretical molecular dynamics study of such clusters. First, the authors investigate the structural variability of the clusters as a function of temperature, and then explore how this variability may affect the potential coalescence of two clusters. The paper is mostly well written and the simulations are performed in a mostly appropriate fashion. I would recommend some methodological clarifications and further work, but I don't think these are very important. However, my main concern with respect to publication at Nature Communications is the level of novelty, or rather the lack thereof. I found the paper a technical paper that does not present any new physics or even unexpected result that would give its readers new intuition on the topic (except some incremental knowledge). As such, I cannot recommend it for publication in Nature Communications.

Author response: We thank the Referee for their positive overall assessment of the work. However, we cordially disagree on the lack of novelty and impact, and have worked to make an extensive revision to highlight those point as well as to address the technical questions raised by all the referees. We note that Referees 2 and 3 regard the work as important, novel, and suitable to be published in Nature Communications.

Briefly, the novelty and impact of this work is based on the following facts:

- Extension of the simulation timescale to 0.1 microsecond range is unprecedented for a cluster of the size Au₁₄₄. This timescale is 4-5 orders of magnitude longer than what

could be achieved by MD simulations using DFT derived forces. Still, our ACE force field reproduces the energetics of the cluster well as compared to DFT (our fig. 2).

- Due to this extended timescale we are able to observe several novel atomistic mechanisms at temperatures 300 – 450 K for a single cluster
- For the first time in our field, we are able to simulate cluster-cluster reaction, illustrating mechanisms by which two clusters can interact exchanging atoms, and how at higher temperature one sees complete coalescence of the clusters into one, larger cluster
- This “reaction product” has a composition very close to a gold-thiolate cluster that has been earlier reported and analysed from experiments. Remarkably, when we optimize the larger cluster, we observe that the atomic structure of the metal core is very similar to the experimentally reported one, based on comparison of calculated and measure PXRD data. This shows that our computational modeling of the cluster-cluster reaction is likely to be very realistic.
- Our work will open doors to further learn about atomistic mechanisms at higher temperatures that are important to understand clusters as catalysts, cluster-cluster reactions, etc.

Hoping that my review will help the authors find a right outlet for their good work, I would also like to add a few questions/comments, which will hopefully help improve the quality of the paper:

1. The abstract does not communicate any particular importance of the work. It should be clearer why this is a high impact paper.

Author response: We have now rewritten the abstract to highlight the importance, novelty and impact of this work (as mentioned above).

2. I believe the Methodology section could be more detailed. For example, what is the statistical ensemble used and how does it reflect the experimental system that is simulated? How were the particles thermalised? In the case of coalescence, were they equilibrated in advance, and, if yes, for how long? Why were the two particular distances between the particles chosen? Since there was a thermostat used, why would the authors expect that a closer distance between clusters affect their degree of coalescence? In my opinion, the authors seem to focus too much on the implementation of the ML potential (which is fine, since this constitutes a methodological advance or novelty), but not enough on the basics of MD.

Author response: We have extended the discussion in the method section by providing more details on the MD simulations. Regarding the questions about the coalescence, both clusters were well equilibrated before they started to interact. The initial distance between the clusters only determines the time scale by how fast the interaction may happen, otherwise it does not have importance here. Temperature of the coalescence simulations is a more important “parameter” – at higher temperatures the clusters have a higher average kinetic energy and reactions over higher energy barrier may take place more easily and in a faster timescale that at low temperatures. This is why we used 550 K as the highest temperature, higher than the maximum 450 K used in single-cluster MD runs.

3. Since velocities are appointed randomly to the atoms in an MD simulation, equilibration is important. Also, statistics. Are the RMSDs of various temperatures reproducible, or are they a result of randomness? Without any statistics we cannot be sure whether the results are physically meaningful. For example, it is not easy to believe that the different features between the 325 and 330 K simulations are down to the 5 K difference. We do have 3 cases for 450 K; why not for the other temperatures? I do not think that 3 cases are enough, by the way, especially considering that they gave so different results. The simulation setup is basic and the authors could easily run a great number of simulations without too much effort.

Author response: Here we took a major task and performed extensive additional MD runs to work out more details and improve the statistics. At each temperature in the single-cluster MD runs, we now have data collected from four parallel, independent “replica” simulations, each spanning the same physical timescale of 120 ns. The new data is visualized in Figure 4 which shows at each temperature the average behavior of the RMSD from those four replica simulations. We note that the extra simulations took a total GPU time of more than two weeks at one of the Finnish national supercomputers so the numerical effort was rather large taking into account also the actual wallclock/waiting time in the queue system. The ACE force field for this system is 4-5 orders of magnitude faster than calculating the interactions “on the fly” from DFT, but still it is much slower than simple classical pairwise force fields (such as Lennard-Jones, Morse,) routinely used in atomistic simulations. One can always discuss “what is a long enough” MD simulation, since the energy landscape for a system like a cluster of 144 Au atoms and 60 thiolate ligand is extremely complex – the number of degrees of freedom is 1332 which is also the dimensionality of the energy phase space. It has become convenient in the MD community to perform parallel, independent “replica” simulations for shorter physical times, instead of one extremely long simulation, since the replica simulations explore different areas of the energy phase space due to different initial velocity distribution and anharmonic nature of the force field. In fact, early fundamental studies of the MD technique showed that two simulations quickly differ in their phase-space trajectories even when started from the same atomic configuration but using different initial velocity distributions, as discussed, e.g., in the classic book of Allen&Tildesly (1987). Hence, replica simulations produce a better statistical sampling in a shorter waiting (wall-clock) time as compared to one extremely long simulation. In the simulations for this revision, we chose to perform four replicas of 120 ns at temperatures of 300 K, 325 K, 350 K, 375 K, 400 K, 450 K. This temperature range is relevant for experiments where thermal effects on the clusters’ atomic structure is expected play a major role for the cluster’s functionality, a typical example being heterogeneous catalysis where clusters are deposited on oxide surface and the system is usually heated up to these temperatures to “activate” the catalyst before cooling it down to the operational temperature (see, e.g., our ref. 36). Figure 4 shows now clearly how the thermal disorder proceeds in a layer-by-layer fashion at range of 300 – 350 K. At 400 -> K one sees more continuous, “diffusive” behavior over all layers, augmented by the big jumps in RMDS caused by the partially or fully detached gold-thiolate fragments that start to appear around 375 K.

4. I am not 100% convinced with the analysis of the RMSD results. RMSD take averages for all atoms of each layer; therefore, if a protrusion is formed (or, especially when a part of the cluster is detached) that totally upsets the averaged RMSD values. I do not see the "migration between

layers" the authors discuss about. For example, in lines 190-93: at 350 K, L2 crosses 0.8 Å and L3 approaches 1.5 Å, but I don't see any line mixing in Fig. 4. In my opinion, all features in Fig. 4 are related to protrusions, not atomic rearrangement.

Author response: See our response above. We have also replotted Figure 4 with the improved data and have extensively revised the discussion of interpretation of RMSD data. For the specific point raised by the Referee, we respond by attaching the structure figures below, showing clearly migration of gold atoms between the layers in one of the runs at 350 K.

I hope that the authors will take the time to work on these issues for the benefit of their already good work so far, and that my suggestions will help them publish a very good paper in another journal.

Author response: We thank the Referee for their critical but constructive comments.

Reviewer #2 (Remarks to the Author):

Metal clusters (NCs) have attracted extensive interest in recent years. The authors selected one of the most studied, near critical sized Au nanoclusters- Au₁₄₄(SR)₆₀ for thermal dynamics investigation at 300 K - 550 K by applying a recently developed machine-learned atomistic potential (ACE). They found that the temperature-induced disordering of the cluster takes place via layer-by-layer mechanism and the outermost gold-thiolate layer, consisting of 30 RS-Au-SR units, spontaneously forms open (polymer-like) or closed (ring-like) gold-thiolate moieties. These fragments may detach and again join the cluster in a dynamic fashion. They also investigated interactions and a reaction between two Au₁₄₄(SR)₆₀ clusters: at elevated temperatures, clusters can exchange gold atoms and thiulates by first establishing a cluster-cluster bridge from a long gold-thiolate polymer; when at close enough contact, this interaction can lead to a complete coalescence of the cluster to form a new roughly spherical cluster. Overall, these findings and results are very interesting, novel and inspiring, which are mirrored by some previous experiments, therefore I can recommend the publication of this nice work after addressing the following issues:

Author response: We thank the Referee for their positive assessment of our work. We do agree that our simulations with the machine-learned ACE potential reveal several interesting novel processes in the atomistic mechanism by how a single cluster disorders at high temperatures and by how two such clusters can react to form one larger merged particle. The potential impact of such results is discussed in our response to Referee 1, and we also rewrote the abstract to highlight the novelty and impact of this work.

1. Why are the durations of 120 ns, 87.27 ns, and 89.85 ns considered in this work? For realistic experiments, they last hours or even days. Does it mean that the simulation can not really reflect the realistic situation?

Author response: As we response to Referee 1, we have now significantly improved the statistical sampling at all temperatures by performing additional MD simulations. It is true that one can never reach even close to the actual timescale of lab experiments – processes like cluster-cluster reactions may indeed take hours to days. At best, we can explore the energy landscape of the clusters up to a few microseconds. This is one reason why we have also performed the cluster-cluster reaction simulation at a rather high temperature, up to 550 K, to speed-up the atomistic dynamics and to facilitate the systems to overcome critical energy barriers for reactions to take place. We also wish to mention that our results should not (yet) be taken literally as describing cluster-cluster reactions that take place in a solvent, since we are not modeling solvent effects at all. Incorporating those would be one natural extension of our work in the future.

2. What is the MD simulation limitations(disadvantages) for metal nanocluster investigations?

Author response: In general terms, MD simulations depend on the quality of the force field used to advance the atomic dynamics. We have verified that the ACE force field used in this work is of good quality as compared to DFT results (see Figure 2 in the manuscript and point 3 below). Another limitation is that even with the most efficient force fields and supercomputers, one can simulate MD trajectories of this kind of systems perhaps only up to a few microseconds (see our point 1 above).

3. Can the authors give a precision and efficiency (time-consuming) comparisons between AIMD method and ML-potential in term of Au₁₄₄(SR)₆₀? Which atomic characteristic does ACE model denote? Why did the authors use this model? How does it accelerate the calculation speed.?

Author response: We base our work on an earlier report (our ref. 15) that describes the ACE model and its fitting to DFT data (structures, forces, energies) of thiolate-protected gold clusters. We also give more details of the model in the Supplementary Information file. The ACE force field is roughly 4-5 orders of magnitude faster than calculating forces “on the fly” from DFT at each atomic configuration (we could think of achieving just a few picosecond physical simulation time from DFT-MD for this system in a supercomputer, instead of about 100 ns with ACE). The accuracy of energies as predicted from ACE vs. DFT energies for the same

configurations was shown in Figure 2 and discussed in that context already in the original manuscript, showing a rather good linear correlation ($R^2 = 0.94$).

4. The RMSD results (Fig.4) can be further clarified or interpreted. For example, it is unknown what is the electronic energy change of each sharp change in Figure 4? Can the energy change of each peak indicate the feasibility of the according reaction? etc.

Author response: We have extensively revised the discussion on Figure 4. The abrupt changes in the RMSD are connected to qualitative changes in the atomic structure, such as formation of polymer like gold-thiolate units or rings that are partially or totally detached from the cluster surface.

5. One of the authors' opinions is that the restructuring of gold nanoclusters starts from the change of the outmost gold-thiolate layer, however, a structure transformation from Au₁₂₇ to Au₁₂₆ was very recently reported, in which the innermost gold of Au₁₂₇ was removed with the left structures including the outmost layer remained after the transformation. Basing on the fact, my question is whether there is also the possibility that the restructuring of gold nanoclusters starts from the innermost gold atom?

Author response: Restructuring of gold clusters (or any metal cluster) can in principle start also from inner parts of the metal core, but it is generally observed (both simulations and experiments) that disordering or structural changes of clusters and nanoparticles start on surfaces. This is simply because atoms or molecular moieties on a nanoparticle surface are less coordinated and have a smaller binding energy than atoms inside the particles, so thermal effects are more likely to break chemical bonds first at surfaces.

Reviewer #3 (Remarks to the Author):

In this work, the authors utilize the atomic cluster expansion (ACE) machine-learning potential to investigate the thermal dynamics and coalescence of Au₁₄₄(SR)₆₀ clusters. While these clusters have been extensively studied within the cluster community, research focusing on their thermal dynamics and coalescence is relatively scarce. This study employs machine learning to extend the time scales that have not previously been explored for Au₁₄₄, unveiling several intriguing findings that pave the way for a deeper understanding of cluster-cluster interactions and coalescence at the atomistic level. I recommend that the manuscript be published in Nature Communications after minor revisions.

Author response: We thank the Referee for their positive assessment of our work.

1. The simulations utilize varying lengths of molecular dynamics (MD). It would be helpful to explain how the decision was made regarding when to terminate the simulations.

Author response: Please see our response to comment 1 of Referee 2.

2. The authors conducted three independent simulations at 450 K. What was the original motivation for this approach, and why do the simulation times differ?

Author response: Running parallel MD simulations for the same system but starting from different initial conditions (initial velocities of atoms) yields an enhanced statistical sampling, as the inherent anharmonic effects of MD force field at higher temperatures lead to chaotic dynamics and drive each simulation to explore different parts of the phase space. In principle, the same could be achieved by simply running one MD simulation long enough times, but in our case where one simulation already takes a few days in a supercomputer, one can get a much more efficient data collection by independent parallel MD simulations (called “replicas” in the MD community) that can be launched at the same time. In the additional work for the revision, we also synchronized the lengths (physical simulation time) of all replicas and run 4 replicas for all temperatures discussed in the paper.

3. The caption for Figure 6 needs to be corrected.

Author response: We thank the Referee for noting the misprint which has now been corrected.

Response to Referees

General remarks from the authors: Referees 2 and 3 now recommend acceptance. Referee 1 has further technical questions but after careful consideration we have decided not to revise the scientific content further, but we provide a point-by-point response below. However, Referee 1 points out a very interesting prior work in the literature that we now added as the new ref. 42 and mention at the end of Discussion. Other content of the MS remains unchanged.

Reviewer #1 (Remarks to the Author):

I would like to thank the authors for heavily revising their original manuscript according to the suggestions by all the reviewers. However, while I appreciate their effort and understand their position, in all honesty, I still do not consider the manuscript fit to publish in Nature Communications. Let me explain:

First, unless I am still missing something important, I find the results rather expectable. In principle, we have two metallic nanoparticles whose ligand shell protects them from coalescing (which would happen spontaneously if unprotected), and when the shell is disrupted by spontaneous fragmentation, the metallic cores can get in touch and the two clusters coalesce. I am not an expert on ligand-protected nanoparticles, but I don't see what is novel here (for example, isn't this, more or less, similar to Bae et al. Nanoletters, 2020, 20, 8704?).

Author response: We thank the Referee for pointing out the nice experimental work of Bae et al. that directly supports our theoretical findings. We now cite this paper at the end of Discussion. That work shows, via TEM imaging, how 2-3 nm gold nanoparticles can merge together to make one larger particle. The theoretical part of that work deals only with classical force field simulations of planar gold surfaces with the ligands used in the experiments, demonstrating the ligand interdigitizing process that “locks” the particles first together when they start to interact. In our work we proved for the first time a full atomistic picture of how coalescence may take place (but with different chemical ligands and interactions as compared to Bae's work), starting from two clusters of **experimentally known size and composition** and producing **a new cluster that also has a known experimental size and composition**. This finding is indeed remarkable.

As I said, I am not an expert on ligand-protected nanoparticles, so let us assume that this result is, indeed, groundbreaking. I am still not sure about how convincing it is. For example, the authors obliged by performing additional simulations. Then, they present in figure 4 the average RMSD evolution of 4 simulations/temperature. Bearing in mind that the individual RMSD graphs from SI are so different to each other, I am not sure there is a point in averaging them. For example, the two spikes around 60 and 80 ns at 375 K come from simulation II, which has a very flat profile otherwise; some protruding chains are created, which soon reunite with the cluster.

The slope in the average profile of figure 4d after 100 ns comes from simulation III, where such spikes were not evident. I don't see the point of averaging.

Author response: Collecting statistics in nanosystems that are on the verge of large structural transitions is always a bit open question. The example case raised by the Referee belongs to T region where significant changes in the structure start to take place, and we observe that in part of the “replica” runs. All runs are still valid paths in the configurational phase space and there is no fundamental reason why one could not average the data.

Interestingly, the SASA probing is not performed for all simulation runs.

Author response: We do not expect to get any additional useful information by doing more of the SASA analysis.

Further, with respect to our previous discussion about diffusion (see Response Letter), I still cannot consider these protrusions "diffusion" of Au atoms to an external shell. The whole structure is totally disrupted, exploding one might say; that is not how I understand diffusion. But, again, perhaps this is just a matter of interpretation.

Author response: Atomic “diffusion” in small particles and at high temperatures is indeed a bit a question of semantics, but we remind the Referee that diffusion takes place also in amorphous materials (disordered systems without long scale order) and we do not see any problem with using this term here. Atomic diffusion effects certainly take place when atoms change layers in the clusters at high temperatures.

Can we be sure that this fragmentation is not an artifact of the inter-atomic potential or of poor relaxation/equilibration? I've had similar effects in similar simulations, and it always turned out I was doing something wrong. This is clearly something very sensitively related to the potential. Has this instability at high temperatures been confirmed by, say, DFT-MD?

Author response: The point in our work is that with the developed ACE ML force field we can perform these dynamical simulations to hundreds of nanosecond timescales. Doing similar simulations with forces calculated from DFT, we could reach perhaps max 10 ps timescale. So, direct verification of the high temperature dynamics is clearly not possible.

Again, with respect to the statistics: the authors claim that when the two coalescing nanoparticles were brought to a higher proximity, they could coalesce (at least more probably than at longer distances). While I don't find this completely implausible, shouldn't this probability depend more on the exact configuration? For example, if two nanoparticles that had a high tendency to fragment were brought together, wouldn't they have a higher probability to coalesce fully? This is yet another claim that needs to be statistically supported further. Can the authors be sure that the same two nanoparticles that coalesced in their simulation would still coalesce if brought together at the same distance but with a different relative orientation?

Author response: Starting the coalescence simulations from various nanoparticle distances would lead to coalescence eventually but with varying physical timescales.

I understand that my comments may be frustrating, after the authors having performed a lot of work, and especially considering that the other two reviewers gave positive assessment. However, I find it hard to accept the manuscript, on grounds of doubts about the conclusions made from the results. I trust the editor can evaluate the situation, and if they find I am being unreasonably doubtful I would be happy to hear the paper was accepted.

Author response: Referees 2 and 3 recommend acceptance and we leave the case now to the hands of the Editor.